# Oriented Crossover in Genetic Algorithms for Computer Networks Optimization

**Furkan Rabee [1]** and **Zahir M. Hussain [1,2,*]**

1 Faculty of Computer Science and Mathematics, University of Kufa, Najaf 540011, Iraq; furqan.rabee@uokufa.edu.iq
2 School of Engineering, Edith Cowan University, Joondalup, WA 6027, Australia
* Correspondence: zmhussain@ieee.org

**Abstract:** Optimization using genetic algorithms (GA) is a well-known strategy in several scientific disciplines. The crossover is an essential operator of the genetic algorithm. It has been an active area of research to develop sustainable forms for this operand. In this work, a new crossover operand is proposed. This operand depends on giving an elicited description for the chromosome with a new structure for alleles of the parents. It is suggested that each allele has two attitudes, one attitude differs contrastingly with the other, and both of them complement the allele. Thus, in case where one attitude is good, the other should be bad. This is suitable for many systems which contain admired parameters and unadmired parameters. The proposed crossover would improve the desired attitudes and dampen the undesired attitudes. The proposed crossover can be achieved in two stages: The first stage is a mating method for both attitudes in one parent to improving one attitude at the expense of the other. The second stage comes after the first improvement stage for mating between different parents. Hence, two concurrent steps for improvement would be applied. Simulation experiments for the system show improvement in the fitness function. The proposed crossover could be helpful in different fields, especially to optimize routing algorithms and network protocols, an application that has been tested as a case study in this work.

**Keywords:** genetic algorithm; uniform crossover; network protocol optimization; routing algorithm; optimization; oriented crossover





## 1. Introduction

A genetic algorithm is a formula for resolving optimization issues that incorporate a constraint and natural selection similar to the biological process that propels evolution. The recent addition of the genetic algorithm (GA) to artificial intelligence was motivated by the biological behavior of chromosomes [1]. The Darwinian evaluation principle known as "survival of the fittest" is what the evolution algorithms do [1]. Therefore, the goal of employing GA is to produce the best offspring (solution), which increases the necessity of adopting it. The GA begins with two parents and mates them to generate new offspring; this mating is termed the "crossover." then the old population is replaced with the new one by using the crossover and mutation operators. This process continues until the convergence condition is met [2].

### 1.1. GA Operands

There are three operands [3] in a typical GA as follows.

1. Selection: This operand determines which chromosomes of the population are selected for reproduction. If a chromosome fits better, it is more likely to be selected for reproduction.
2. Crossover: This operator exchanges the subsequences between two chromosomes before and after a locus that is randomly chosen to create two offspring [3].

3. Mutation: This procedure involves flipping one or more randomly selected bits in the parent's chromosomes to create an offspring from a single parent [4]. Any bit has a slight chance of mutating, like 0.001 [3]. The layout shown in Figure 1 represents the typical GA process.

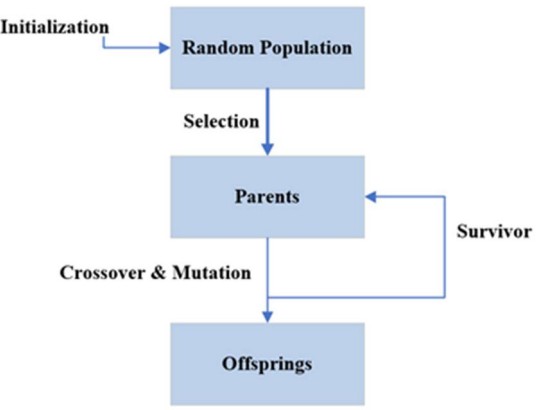

**Figure 1.** Typical genetic algorithm design.

This approach is based on the observation that specific chromosomally encoded traits are shared by individuals and can be passed to )inherited by( their offspring through crossover [5]. The genes of either one parent or both parents, with mutations, are shared by two offspring.

*1.2. Single–Point Crossover*

The best-known and frequently applied crossover model so far among researchers is that presented by Ref. [6]. A crossover site is randomly selected along the length of the matched strings, and bits that are immediately near the cross-sites are exchanged.

The beneficial traits of the parents may be combined to produce better offspring when the right site is chosen. If the right place is selected when good parents are mated, the offspring will be better; if not, the string quality will be severely hampered. If the head and tail of one chromosome contain acceptable genetic material, then no offspring will acquire the two beneficial traits straight after the single-point crossover.

*1.3. N-Point Crossover*

Ref. [7] was the first to use the n-point crossover. It was similar to the single-point crossover. In a two-point crossover, there are two relevant crossing sites. The performance of the genetic algorithms can be severely impacted by interruptions of building blocks caused by the continual addition of the crossover sites.

*1.4. Uniform Crossover*

Ref. [8] presented a uniform crossover, where the chromosomes are not broken up by uniform crossover for recombination. Each gene in a child's offspring is made by copying it from a parent who has been selected based on the bit that corresponds to it in a binary crossover mask that has the same length as the parent chromosomes. The two parents are chosen for crossover through the uniform crossover; it produces two children with n genes uniformly chosen from both parents. A random real integer determines whether the first child chooses the $i^{th}$ gene from the first or second parent [9].

*1.5. Numerical Chromosome Representation*

The crossover mostly used binary encoded chromosomes, and for real-value encoding, the numerical crossover is utilized. Here, two parent chromosomes are combined linearly by the numerical crossover operator. Two chromosomes are randomly chosen for crossover,

resulting in two children who are a linear blend of their parents. N-point is mainly used in the case of binary encoded chromosomes [1].

### 1.6. Operators Definition

Numerous operators represent the main parts of a genetic algorithm form. A gene is a string of bits or a real number within a specific length. A chromosome is a term used to describe a sequence of genes. An allele, which can be represented by a symbol or a bit, is the smallest chromosomal unit. While a phenotype offers an external description of the individual, a genotype is a piece of data contained in a chromosome [4,10]. The main operands of GA are depicted in Figure 2.

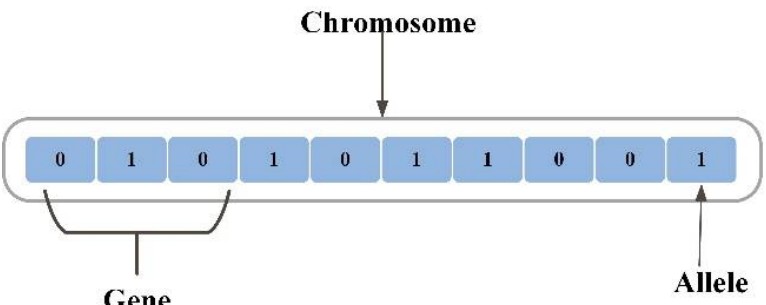

**Figure 2.** The GA operators.

The contribution of this article is a novel chromosome in the GA optimization field, designed by a new construction of the allele, where the new design suggests that each allele in the chromosome contains two attitudes, one good (or preferred) and the other is bad (unpreferred). Then the two new levels of crossover are applied: one before the mating process and the other comes after. This optimization method would improve the good (or preferred) attitude at the cost of the unpreferred one. Thus, the proposed process would improve the good parameters at the cost of the bad parameters in the system. The new optimization method comes from changing a core process inside the genetic algorithm, as we will explain in Sections 4 and 5.

## 2. Related Works

### 2.1. Original Theories

Ref. [8] was first to present the uniform crossover for GA, even at one point, the second point was presented, but the uniform crossover showed it outperform in optimization, and till now, this crossover type is applicable in many different science fields.

Ref. [7] presented an adaptive algorithm to decide when a particular crossover (one point, second point, or uniform) will be optimal for any problem. However, it still works with the standard crossover.

Ref. [9] presented an excellent review showing more than thirty-five types of crossovers presented till 2015, and all the suggested research used in different optimization fields.

Ref. [4] presented a review that shows the importance of GA in the optimization for machine learning and deep learning.

A schema that included a two-point crossover was published in Ref. [2], where the proposed methods offer a contrastive convergence rate.

When the balance between the traits of parents and offspring was a challenge in GA optimization, Ref. [5] presented balanced crossover operators that guarantee the offspring has the same balanced features as the parents.

Ref. [4] presented a modified optimization method that depended on AI, and presented guidance for both beginner and experienced researchers designing evolutionary neural networks, assisting them in selecting appropriate genetic algorithm operator values for use in applications in a certain issue domain.

### 2.2. Implementation of GA

Various research works have been published on GA. The latest works will be discussed in this section.

A genetic algorithm was presented in Ref. [11] to reduce the power losses caused by turbine wakes in wind farms.

A new flight trajectory computation made with GA was proposed by Ref. [12]. The approach examined lateral and vertical navigation to determine the most fuel-efficient cruise trajectory, with optimization by GA saving up to 5.6% in gasoline.

Ref. [13] used deep learning with a convolutional neural network to classify four types of leucocytes. A genetic algorithm was used to optimize the CNN's hyperparameters; this article showed that CNN is not efficient in getting optimal performance.

Ref. [14] presented a new optimization method called the puzzle optimization algorithm (POA), which can be used in different optimization problems. The advantage of this method is that there are no control parameters, thus not requiring parameter settings.

Ref. [15] presented a technique based on fuzzy triangular numbers that have been applied to simulate the recruitment process of the individual to the employee. Moreover, a genetic algorithm has been used for optimization, where the author presented a solution for the selection process to the best individual through a GA and fuzzy ranking.

Ref. [16] used ensemble classifiers to present a student predictive model and pre-pressing to implement a search before classifying via data-mining methodology in a context of educational data-mining (EDM). The best solution was then discovered, and GA was utilized to look for issues and raise the likelihood of reaching a solution.

Ref. [17] offered an innovative way in strong-rule generation, one of the key components of data-mining, where the author employed it differently from the existing construct rules. However, the best optimization technique for generating rules was the genetic algorithm.

Ref. [18] tried to determine the optimal choice for scheduling generator maintenance, where various multi-objective optimization methods were examined. One optimization method, non-dominated sorting genetic algorithm, exemplifies the importance of utilizing GA and optimization in a variety of scientific fields.

Ref. [19] presented a new crossover operand for the genetic algorithm called quarterly crossover (QA), by assuming a different structure for genes within the chromosome, which resulted in two crossovers intended to be an optimized solution for real-time scheduling. This reference presented expanded optimization tools and parameters for the network.

Ref. [20] presented an optimization method for an ad hoc network to be used for roadside units (RSUs) with the expectation that the proposed optimization would reduce accidents and traffic jams, but this method missed some factors related to GA and required more factors for optimization.

Ref. [21] presented a model for schedule risk management of IT outsourcing by using the distributed decision making (DDM) theory and the principal-agent theory as well as designing a hybrid algorithm from the genetic algorithm and simulated annealing algorithm. The designed algorithm focused on the risk management problem. This article is aimed at giving the decision-maker the scientific quantitative tool they need to manage the schedule risk of an IT outsourcing project.

Ref. [22] presented the bit masking oriented genetic algorithm (BMOGA) for context-free grammar induction. It used a Boolean-based approach divided into two stages, utilizing the advantages of crossover and mutation mask-fill operators to direct the search process from the $i^{\text{th}}$ generation to the $(i + 1)^{\text{th}}$ generation. To produce an appropriate amount of population in each generation, crossover and mutation mask-fill techniques are used. The article focused on the grammar of the context induction (as opposed to our current work, which will focus on the computer network optimization).

In Ref. [23], the principal-agent theory was applied to ITO projects to reduce schedule risk. For the purpose of describing the vendor and client decision-making process, a two-level mathematical model was constructed. The size of the problem grows significantly as

the number of activities and subprojects rises. The resulting optimization is a continuous-domain, NP-hard problem. A genetic algorithm (GA) is presented to solve this problem, where the GA model used strong optimization abilities for convergence, reliability, and efficiency, which is a good tool for this kind of optimization problem (as opposed to our proposed crossover that can be applied for optimization in various fields, especially computer networks).

In the field of metaheuristic optimization, Ref. [24] presented a hybrid metaheuristic algorithm for a location-routing problem (LRP), tackles facility location problems and vehicle routing problems simultaneously to obtain the overall optimization. This article did not handle the GA crossover and did not use it (in contrast to our proposed work, which modified the core of the GA crossover to get a novel adaptive crossover).

Ref. [25] is a recent article, where GA has been used in the smart contract of block chain technique to predict a new offspring of the animals endangered. This article presented a decentralized application (SONR DAPPs) by implementing a genetic algorithm to forecast a brand-new offspring with improved characteristics.

From the above recent works, we found that GA has applications in different fields, thus we expect that the proposed work will affect various fields by solving a drawback of the existing GA structure.

### 2.3. Optimization in Fields of Science

The field of cloud computing improvement and WSN is getting a lot of attention among other fields, where the energy consumption of nodes in the network should be restricted. Therefore, Ref. [26] used an optimization technique to reduce the utilization of the energy by the data center in a way to accomplish the best quality of service.

Refs. [27–29] presented an energy-efficient deployment method for sensor nodes by clustering the nodes and applying an optimization technique for the optimal reduction of the nodes' battery. The optimization for nodes led to the optimization of the routing protocol. Ref. [30] used a GA for mechanical engineering in multi-scale surface roughness, and presented three genetic algorithms to superimpose and merge the mathematical descriptions of chromosomes to determine the best roughness features.

## 3. Area of the Proposed Work

The main goal of network optimization is to reduce the problems that occur in any network, to get the best performance at the lowest possible cost. The network must promote increased throughput and usability, while allowing data to flow effectively and efficiently. This is accomplished by managing network latency, traffic volume, network bandwidth, and traffic direction [31].

### 3.1. Applications of the Proposed Work

The proposed work can be utilized in many applications affected by the field of optimization, such as computer networks, routing algorithms, wireless sensor networks, etc. Our focus in this paper will be routing algorithms, where efficient routing of packets through nodes is very important because it will find the path between two entities (two nodes) with the least amount of cost and disturbance [32]. The proposed modification to the genetic algorithm optimizes the distance since the proposed method (as opposed to existing approaches) takes into account network congestion when optimizing.

### 3.2. Minimal-Cost Network Flow

The minimal-cost network-flow problem deals with a single commodity that must be distributed over a network [33]. Suppose there is a network with five nodes, as shown in Figure 3; the minimal-cost network-flow problem deals with a single commodity that must be distributed over a network. Consider a directed graph $G = (N,A)$ with the set of nodes $N$ and the set of links A. The cost of sending a unit flow on link $(i,j) \in A$ is $c_{ij}$, also $x_{ij}$, which is the amount of flow on link $(i,j)$.

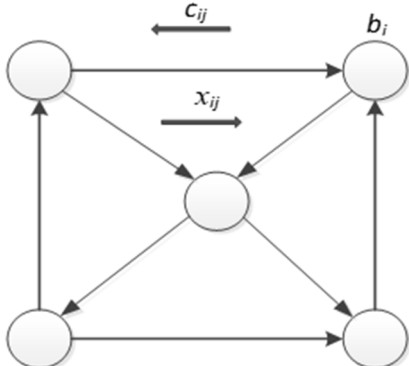

**Figure 3.** Distributed nodes example.

We first define $b_i \forall i \in N$, where $b_i$ denotes the amount of supply for source nodes and $b_i > 0$, on the other hand $b_i < 0$ denotes the demand size for sink nodes. Also, $b_i = 0$ will be for intermediate nodes. For simplicity, we assume the $\sum_{i \in N} b_i = 0$, which can be relaxed easily.

The minimal cost of such a problem is to optimize Equation (1) [33].

$$z = \sum_{(i,j) \in A} c_{ij} x_{ij} \tag{1}$$

There may be upper bounds on $x_{ij}$, denoted by $u_{ij}$; that is, $x_{ij} \leq u_{ij}$. If there is no upper bound on link $(i,j)$, the constant $u_{ij}$ shall be set to infinity.

## 4. System Setup and Main Definitions

Suppose a population P with N persons, where each chromosome has Z genies, each gene contains n alleles, and every one of them has two properties, A and B. Let A and B refer to the two attitudes for any system. If this system prefers one of these attitudes to be higher than the other, then we will say that A refers to a positive attitude and B to a negative one, as shown in Equations (2) and (3).

$$A_T^x = \sum_{i=1}^{n} A_i \tag{2}$$

$$B_T^x = \sum_{i=1}^{n} B_i \tag{3}$$

where $x$ refers to the specific person and $n$ refers to the total number of alleles, $i$ refers to individual number of allele. The relationship between $A$ and $B$ is an inverse relationship. It means as $A$ increases, $B$ should be decreased. In fact, $A$ is the complement of $B$, as per Equation (4):

$$A_i + B_i = 1 \forall_{n \in x} \tag{4}$$

The structure of the proposed chromosome is presented in Figure 4, where each allele has been represented by $A$ and $B$.

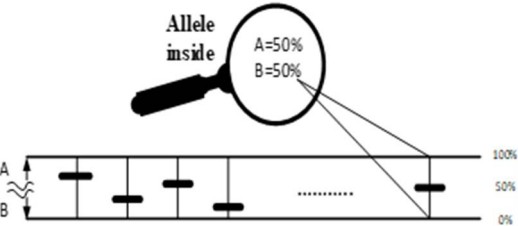

**Figure 4.** The proposed chromosome.

These properties can be affected by mutation ($\mu$) [34,35], positive mutation ($\gamma$), and negative ($\delta$) mutation, where $\gamma$ increases the positive attitude (A). Conversely, the negative attitude (B) will decrease according to Equation (4) and vice versa. The following algorithm shows the mutation process affecting the allele (A and B). Suppose that the mutation ($\mu$) comes randomly (positive or negative), as follows:

$$\mu = \begin{cases} \gamma + ve \\ \delta, -ve \end{cases} \tag{5}$$

$$\gamma = \{\gamma_1, \gamma_2 \ldots \ldots, \gamma_n\} \tag{6}$$

$$\delta = \{\delta_1, \delta_2 \ldots \ldots, \delta_n\} \tag{7}$$

The value (weight) for each mutation should be minimal, about $\frac{\mu}{1000}$, especially with numerical crossover. So, the change would be small with each crossover.

**Step 1**: Start
**Step 2**: Initialize each n population of chromosomes randomly
**Step 3**: Generate random mutation ($\mu$ )
**Step 4**: If $\mu$ is $\gamma$ = +ve

A is increased and B decreased

**Step 5**: Else If $\mu$ is $\delta$ = -ve

A is decreased and B increased

**Step 6**: $A_i + B_i$
**Step 7**: End.

## 5. Oriented Crossover (OC)

This system is supposed to enhance the allele by improving one attitude at the expense of the other by applying the mutation weight. As the first step, this crossover happens for the single parent, and the effect of $\mu$ will make a change for the alleles. If this crossover could not improve the system (or reach the target), the second crossover applies to mate the parents. Figure 5 shows the oriented crossover structure. The proposed crossover can be applied to the binary crossover and numerical crossover.

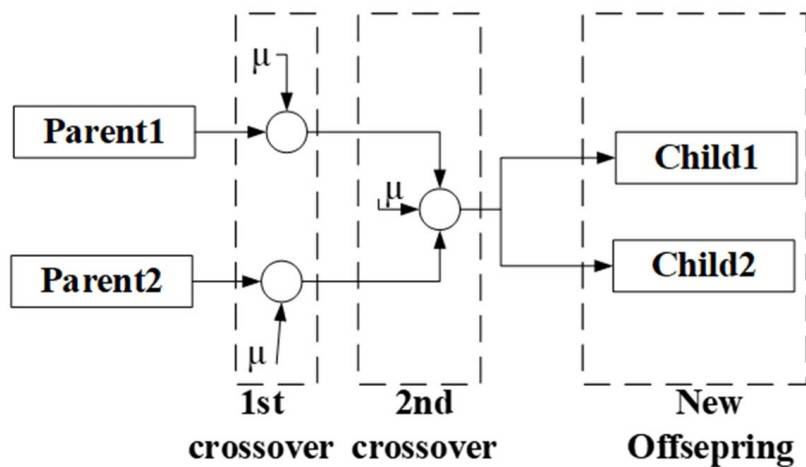

**Figure 5.** Oriented crossover structure.

### 5.1. First Crossover

This type of crossover could be deployed in two modes depending on the type of application: Type-1 is with the parameters dependent on binary facts, while the Type-2 application depends on numerical values and factors. The types are as follows:

#### 5.1.1. Binary Crossover

We can represent this crossover for *A* and *B* attitudes to obtain a binary form for each gene by Equations (8) and (9).

$$A_{nn}^{P} = \begin{bmatrix} A_{11}^{p} \middle| \gamma_{11} \\ A_{22}^{p} \middle| \gamma_{22} \\ \ldots \\ \ldots \\ A_{nn}^{p} \middle| \gamma_{nn} \end{bmatrix} \tag{8}$$

$$B_{nn}^{P} = \begin{bmatrix} B_{11}^{p} \middle| \delta_{11} \\ B_{22}^{p} \middle| \delta_{22} \\ \ldots \\ \ldots \\ B_{nn}^{p} \middle| \delta_{nn} \end{bmatrix} \tag{9}$$

After implementing Equations (8) and (9), the new attitudes should be getting from the same single parent, then we need to add logically (OR) both attitudes to get a newly single parent, as in Equation (10):

$$P_{n}^{S} = \begin{bmatrix} A_{11}^{P} \middle| B_{11}^{P} \\ A_{22}^{P} \middle| B_{22}^{P} \\ \ldots \\ A_{nn}^{P} \middle| B_{nn}^{P} \end{bmatrix} \tag{10}$$

where $\mu = \begin{cases} 1 \equiv \delta \\ 0 \equiv \gamma \end{cases}$, *P*: new parent, p: particular individual for the same parent.

#### 5.1.2. Numerical Crossover

This type of crossover works with finite values for *A*, *B*, and μ. The weight μ has three levels, as in Equation (11).

$$\mu|_{\forall_{\gamma,\delta}} = \begin{cases} \text{High} \\ \text{median} \\ \text{low} \end{cases} \tag{11}$$

where μ is normally very small.

The crossover for the numerical crossover obtained by two cases depending on the state of μ (+ve or –ve), as follows:

In case μ comes in +ve state, we have:

$$A_{nn}^{P} = \begin{bmatrix} A_{11}(1 + \gamma_1) \\ A_{22}(1 + \gamma_2) \\ \ldots \\ \ldots \\ A_{nn}(1 + \gamma_n) \end{bmatrix} \tag{12}$$

$$B_{nn}^P = \begin{bmatrix} B_{11}(1 - \gamma_1) \\ B_{22}(1 - \gamma_2) \\ \dots \\ \dots \\ B_{nn}(1 - \gamma_n) \end{bmatrix} \tag{13}$$

In the case μ comes in the negative (-ve) state, we have:

$$A_{nn}^P = \begin{bmatrix} A_{11} - B_{11} \times \delta_1 \\ A_{22} - B_{22} \times \delta_2 \\ \dots \\ \dots \\ A_{nn} - B_{nn} \times \delta_n \end{bmatrix} \tag{14}$$

$$B_{nn}^P = \begin{bmatrix} B_{11}(1 + \delta_1) \\ B_{22}(1 + \delta_2) \\ \dots \\ \dots \\ B_{nn}(1 + \delta_n) \end{bmatrix} \tag{15}$$

$$P_{nn}^P = \begin{bmatrix} A_{11}^{new} + B_{11}^{new} \\ A_{22}^{new} + B_{22}^{new} \\ \dots \\ \dots \\ A_{nn}^{new} + B_{nn}^{new} \end{bmatrix} \tag{16}$$

*5.2. Second Crossover*

We can call this type of crossover a mate crossover because this crossover between the parents comes after the first crossover. This crossover has been represented in Equations (17) and (18).

$$P_{1i}^{OFS} = \mu_i P_{1i}^p + (1 - \mu_i) P_{2i}^p \tag{17}$$

$$P_{2i}^{OFS} = \mu_i P_{2i}^p + (1 - \mu_i) P_{1i}^p \tag{18}$$

where $P_{1i}^{OFS}$ and $P_{2i}^{OFS}$ refer to new offspring for the corresponding parent.

Second crossover is equivalent to uniform crossover [1].

## 6. Experimental Designs

This section presents the implementation details of the proposed system. The proposed algorithms are used as optimization methods.

*6.1. Optimization Procedure*

Here, we show the procedure of implementing the required populations.

**Step 1:** Initialization for population chromosomes starts by generating random individuals as initialization, each individual's chromosome with ten "*A*" attitude alleles for each, as proposed in Section 4.

**Step 2:** The range of *A* is [0, 1] for binary crossover, and from 0.1 to 0.9 for each allele in the numerical crossover.

**Step 3:** Take the complements of *A* for each allele to generate the B allele, as discussed in Section 4 of this article.

**Step 4:** Calculating the cost function for each parent depends on which equation that needs to be optimized.

**Step 5**: Generate and test four isolated populations using Equations (8)–(11) for binary crossover, and a population for numerical crossover using Equations (17) and (18).

**Step 6**: Apply the optimization in Equations (19)–(22) as a cost function for each parent in the population with each scenario.

*6.2. General Optimization Test*

Two approaches have been used for general mathematical optimization as follows:

- Uneven Decreasing Maxima Function [2,36,37]: this is one of the multimodal optimization problems:

$$F = e^{\left(-2\log(2)\left(\frac{\varnothing - 0.08}{0.854}\right)^2\right)} \sin^6\left(5\pi\left(\varnothing^{\frac{3}{4}} - 0.05\right)\right) \tag{19}$$

where $\varnothing \in [0,1]$. The aim is to obtain an optimal value of $\varnothing$.

- Himmelblau Function: this function, which is used by [2,31], is defined as:

$$F = 200 - \left(\varnothing_1^2 + \varnothing_2 - 11\right)^2 - \left(\varnothing_1 + \varnothing_2^2 - 7\right)^2 \tag{20}$$

where $\varnothing_i \in [-6,6]$. The aim is to obtain optimal values of $\varnothing_1, \varnothing_2$.

*6.3. Communication and Network Optimization*

The main equations that have been used in communication and network optimization are as follows:

$$F = \sum_{i=1}^n x_i y_i \tag{21}$$

$$F = \sum_{i=1}^n (x_i y_i)^2 \tag{22}$$

$$G_T = \sum_{i=1}^n C_i^p \tag{23}$$

where $n$ is the number of individuals in that generation, $C_i^p$ refers to the individual $i$ whose cost function is $p$.

The optimization process in this section is performed by the following steps:

**Step 1:** Optimize the parameters in computer networks as per Equations (21) and (22), which leads to minimizing the value of $F$.

**Step 2:** Choose ten individuals to represent the first generation and then calculate the fitness of this generation or Generation Fitness ($G_T$) as per Equation (23).

The fitness is calculated by applying Equations (19)–(23), then calculating the fitness of each individual as per Equation (24).

$$f_i^p = \frac{C_i^p}{G_T} \tag{24}$$

**Step 3**: Calculate the probability for each individual as per Equation (25).

$$Prob_i = f_i^p * n \tag{25}$$

Then the individuals are rearranged depending on their probabilities above for the purpose of choosing the best individual to be the parent of this generation.

**Step 4**: After arranging the individuals in decreasing order based on their probabilities, eliminate the last two individuals (the two with the least probabilities).

**Step 5**: After eliminating two individuals, it now has the best parents ready to mate; thus, it makes the crossover according to the proposed OC algorithm to get new offspring as a new generation.

**Step 6**: Repeat the steps from (2) to (6) again till they reach only two individuals in the offspring.

**Step 7:** Repeat steps (1) to (7) for one hundred iterations to choose optimal values.
**Step 8:** The steps from (1) to (8) are implemented using Equations (19)–(22).

### 6.4. Fractal-Based WSN Optimization

In this section, the proposed method is utilized in the fractal-based design of wireless sensor networks (WSN) for optimization. This fractal design is based on geometrically-based patterns and structures that can repeat in any size, from the largest shape to the symmetric smallest shape. The Sierpinski triangle [38,39] is a well-known fractal model. Figure 6 shows WSN nodes based on the Sierpinski triangle.

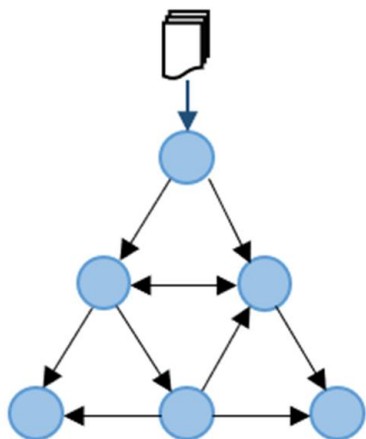

**Figure 6.** Fractal-based WSN nodes.

In Figure 6, it is supposed that the nodes are deployed as the Sierpinski triangle fractal, for static task (message) scheduling. The purpose of optimization is to reduce the overhead on the nodes. The overhead increases when the message is delivered to the base of the triangle. The Sierpinski triangle nodes topology is constructed under two formulas:

- General formula for nodes numbers is:

$$N = 3 * 2^k \tag{26}$$

where $N$ is the number of nodes, and $k = 0, 1, 2, \ldots, m$.

- General formula for the number of links (edges) is:

$$L = (N * d)/2 \tag{27}$$

where $d$ is the degree of node.

Equation (27) will be used in Section 7.6 of this article.

### 6.5. Recursive Process for Fitness

The experimental work involves numerous recursive processes, a fact which implies that each generation's output will serve as the next generation's input. However, it will depend on which crossover will be selected as the next generation's input. The recursive process for generations depends on their fitness. Figure 7 shows the flow work of the recursive processes.

In this article, the performance was tested for five generations. Figure 7 represents the parent selection process to produce a new generation based on fitness values. The blue ball refers to the initialization of fitness values as produced by all types of crossover methods. The output consists of two balls, red and green, where "red" represents the output from the implementation of the uniform crossover (UC) methods, and the "green" represents the output from the implementation of the OC crossover methods.

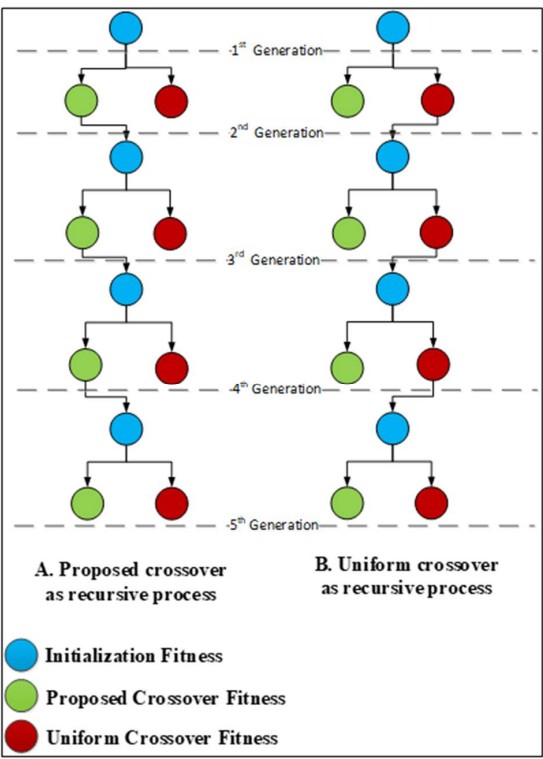

**Figure 7.** The recursive processes for fitness.

Figure 7A shows the OC fitness as a recursive fitness input for the next generation. Figure 7B shows the UC fitness as a recursive fitness input for the next generation. We used different recursive methods to show the influence of the proposed method. The proposed method shows best performance in both of the above recursive methods. The steps in Sections 6.1–6.3 have been implemented using the UC and OC crossover with each generation.

### 6.6. Implementation Phases

The experimental work was performed in two phases as follows:

- Phase one: this phase implements the proposed optimization approach with one type of crossover, which is UC.
- Phase two: this phase implements the proposed optimization comparatively using UC, NC (with different values of N), and QA [19].

## 7. Results and Discussion

### 7.1. Genetic Algorithm Parameters

We have carried out various experiments to implement the proposed approach in MATLAB (under academic license 40635944). Table 1 shows the GA parameters used in this work. Different crossovers will be tested and compared.

**Table 1.** Genetic algorithm parameters.

| Parameters | Value per Population | Factor |
|---|---|---|
| Population Size | 10 | ×12 |
| Scaling Function for selection probability | Uniform distribution | … … … … |
| Selection Operator | Roulette Wheel | … … … … |
| Crossover Probability | 80% | ×12 |
| Mutation Operator | OR | … … … … |
| Mutation Probability | 10% | … … … … |

### 7.2. Binary Crossover under Phase One

Here, we evaluate the performance of the binary crossover for the chromosome contents. The cost function for each individual has been calculated after converting the binary into decimal values. The binary crossover was implemented for five generations to test the four Equations (19)–(22).

The criterion of comparison between algorithms is the population's fitness for each generation. We implemented the proposed crossover and the standard uniform crossover. The fitness comparisons are shown in Figures 8–11. The original population fitness (without applying any algorithm) is called "Initial", and this population is in need to be optimized by applying UC or OC. Figure 8 shows the implemented of Equation (19); where it is evident that OC outperforms UC. Figure 9 shows the implementation of Equation (20); noting that OC outperforms others in the last generation. For Figures 10 and 11, the OC worked as well as UC (in the fourth generation).

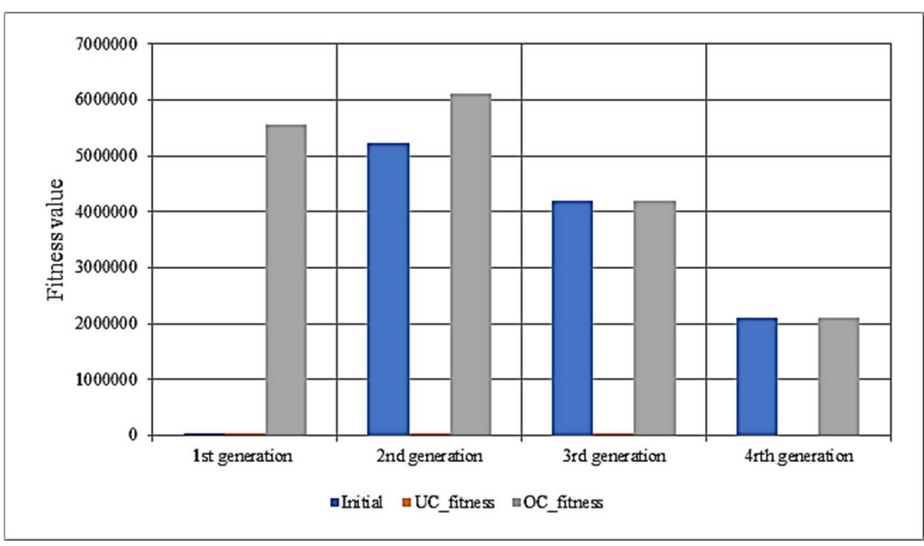

**Figure 8.** Fitness comparison under binary crossover using Equation (19).

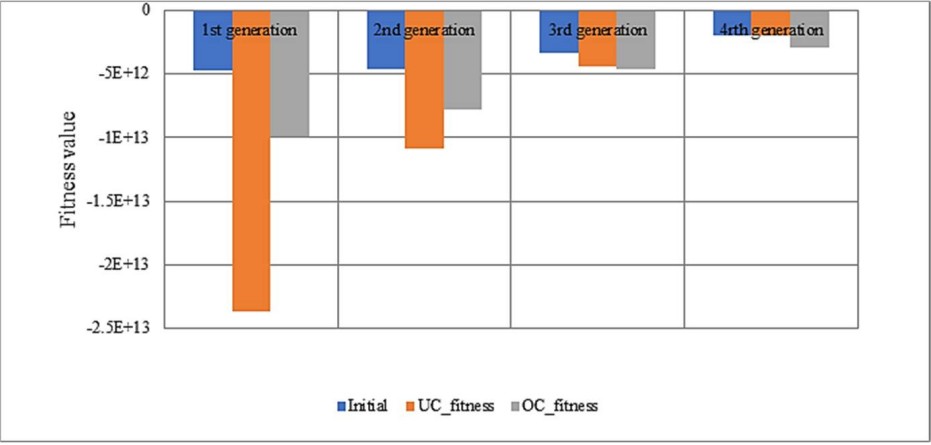

**Figure 9.** Fitness comparison under binary crossover using Equation (20).

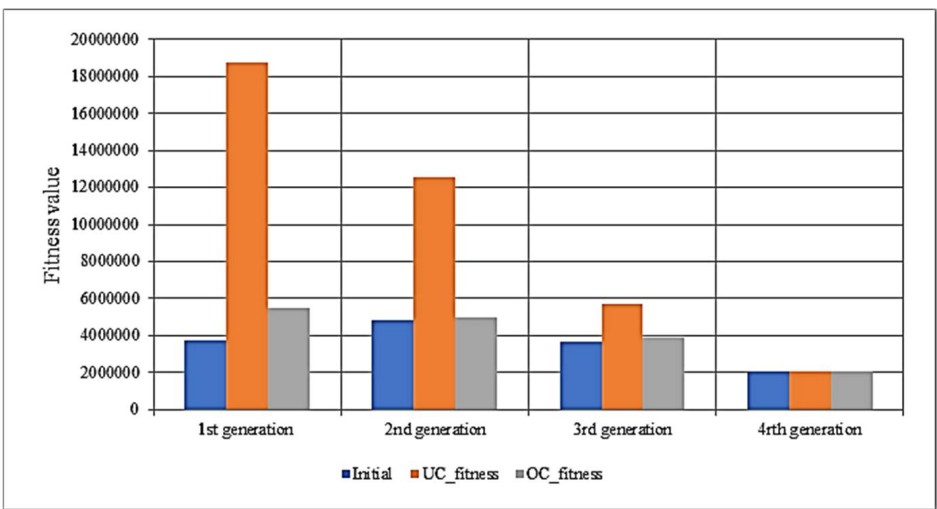

**Figure 10.** Fitness comparison under binary crossover using Equation (21).

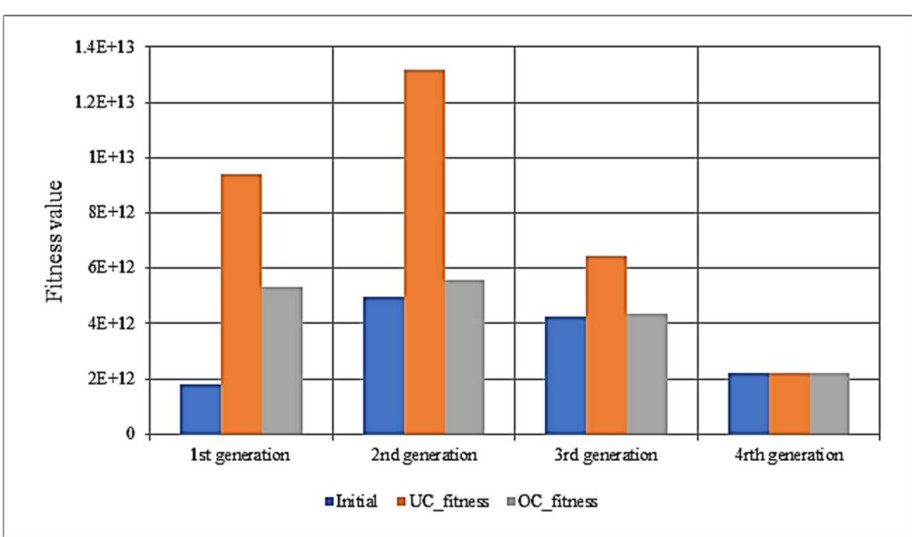

**Figure 11.** Fitness comparison under binary crossover using Equation (22).

### 7.3. Numerical Crossover under Phase One

With the numerical crossover, the floating points representing *A* or *B* attitudes are used randomly (range from 0.1 to 0.9). For this type of crossover, the following equation is used as the cost function:

$$F = \sum_{i=0}^{n} z^i \tag{28}$$

where $z$ refers to the parameters in Equation (1) that are used to calculate the fitness for each link *i*. This function is used for communication and networking applications.

The system was implemented for Initial, UC, and OC fitness functions. Figure 12 shows fitness comparison, where the proposed OC optimization has a clear effect.

### 7.4. Equilibrium State under Phase One

In this section, a comparative search for the best optimization method after one hundred iterations, where every iteration involves four generations from the selected population. Figure 13 shows the counts of iterations to reach the equilibrium state using Equation (19). Figure 14 shows the equilibrium state using Equation (21), while the equilibrium using Equation (22) is shown in Figure 15. For Equation (20), the equilibrium states are similar for all algorithms.

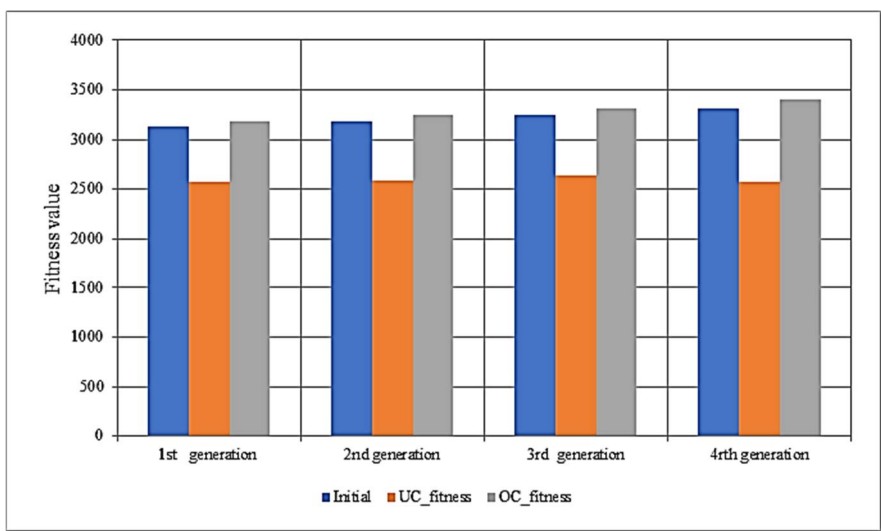

**Figure 12.** Fitness comparison under Numerical crossover.

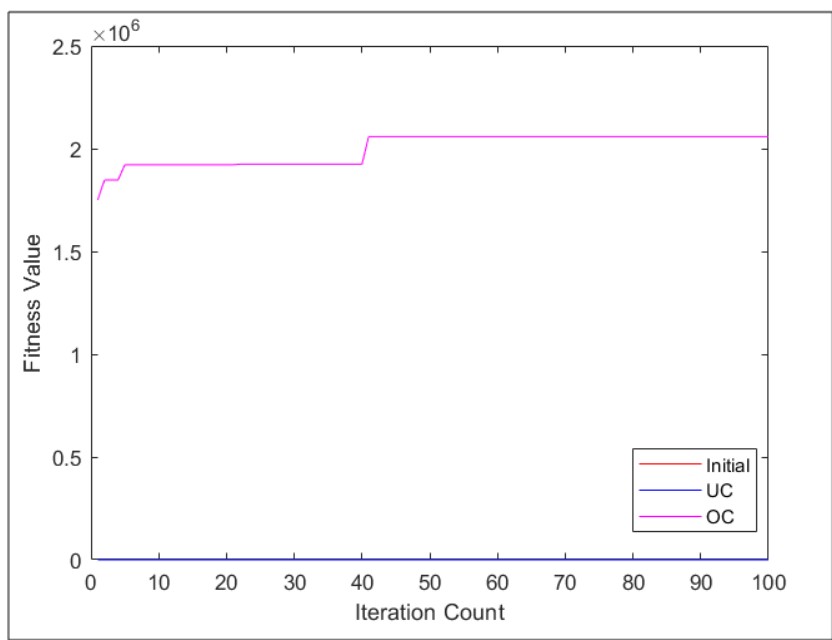

**Figure 13.** Equilibrium states using Equation (19).

*7.5. Binary Crossover under Phase Two*

In this sub-section, we assess the chromosome contents using the binary crossover. The cost function for each individual is determined by converting the binary values to decimal. The binary crossover technique is tested for four Equations (19)–(22) over four generations. The performance of different algorithms is compared based on the fitness of the population for each generation. An experimental evaluation is conducted on the proposed OC and the existing UC, NC, and QA. The fitness evaluations are illustrated in Figures 16–19, which depict comparisons between the initial population fitness and the fitness of populations that have undergone optimization using UC, NC, QA and OC. Figure 16 shows the implementation of Equation (19); it is evident that OC gradually reduces the fitness for its population across generations, outperforming other algorithms which exhibit constant behavior. Figure 17 shows the fineness of Equation (19) by Japan's candles, where only OC appears as a dynamic candle. Figure 18 demonstrates the implementation of Equation (20): although QA approaches the maximum first, such maximization may not be suitable for all systems as it was suitable for a specific system with low range (min. to max.) optimization,

and such behavior is not suitable in many applications. Elsewhere, OC is gradually maximized, and it seems to be suitable for different systems. Figure 19 shows clearly that OC has a gradual optimization behavior and good performance with standard crossovers, in contrast to QA, which looks like handling a special case of optimization.

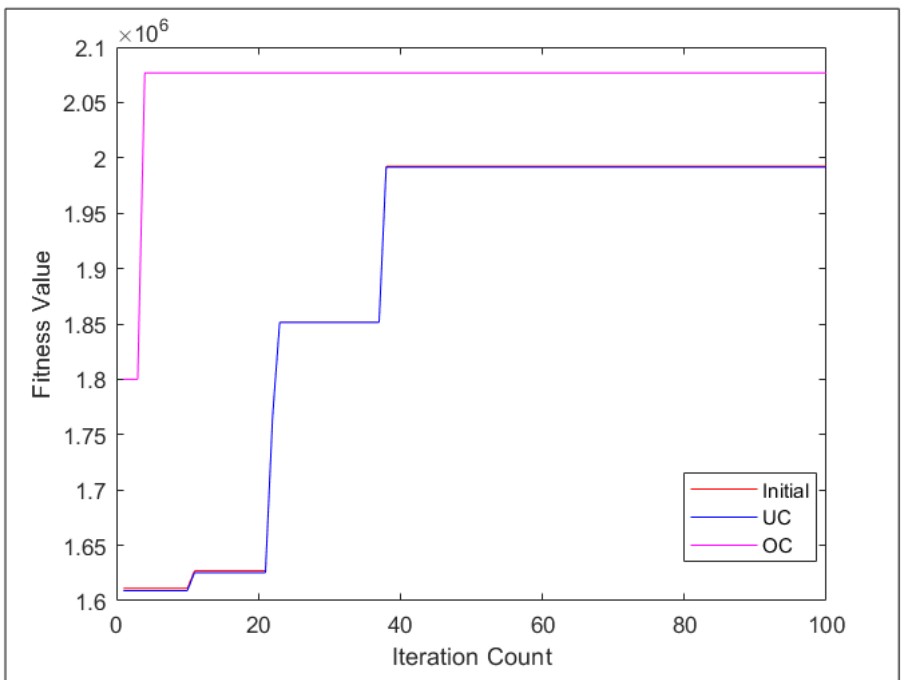

**Figure 14.** Equilibrium states using Equation (21).

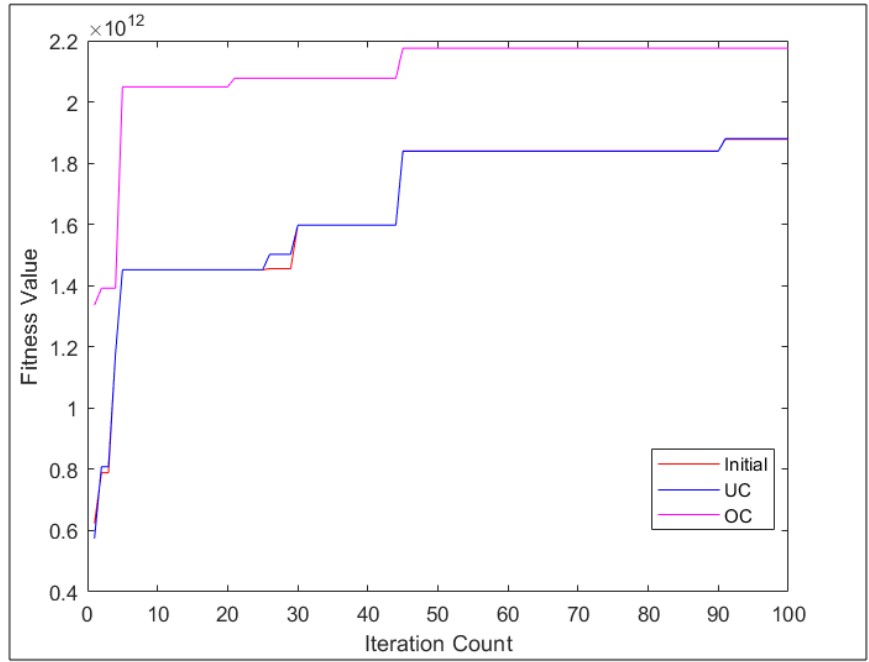

**Figure 15.** Equilibrium states using Equation (22).

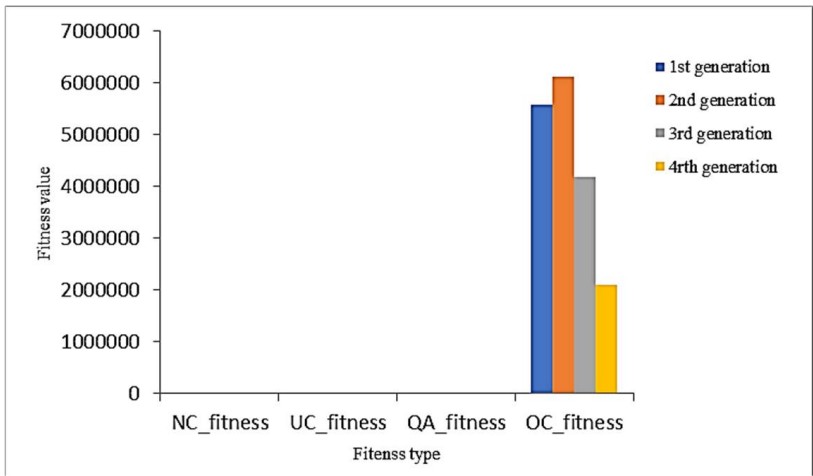

**Figure 16.** Fitness comparison of crossover's types under binary crossover using Equation (19) for four generations.

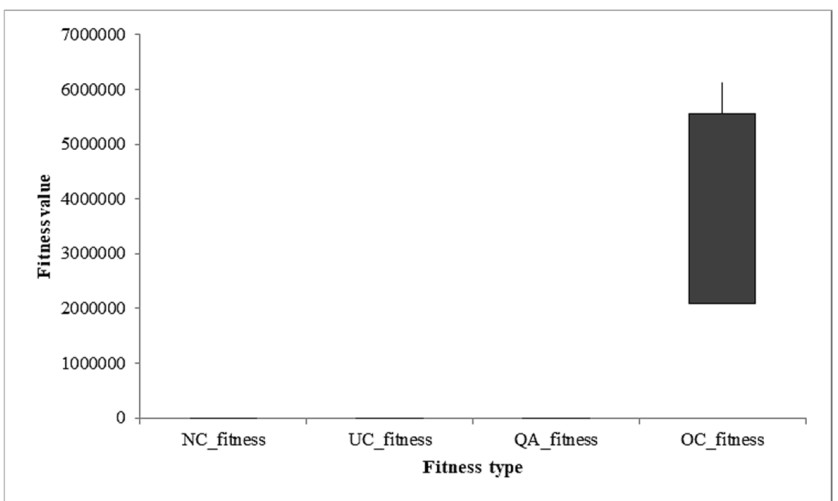

**Figure 17.** Fitness comparison of crossover's types under binary crossover using Equation (19) sampled by Japan's Candles. Black color for candles refers to decreasing in value.

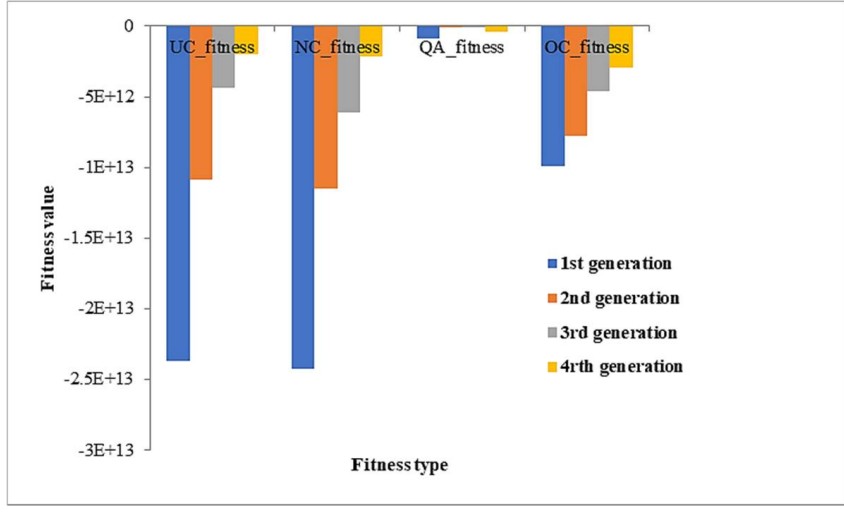

**Figure 18.** Fitness comparison of crossover's types under binary crossover using Equation (20).

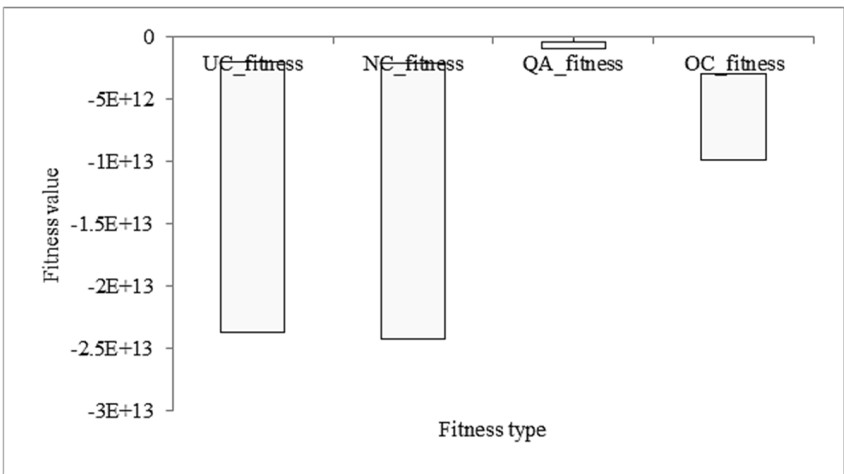

**Figure 19.** Fitness comparison of crossover's types under binary crossover using Equation (20) sampled by Japan's Candles. White color for candles refers to increasing in value.

*7.6. Numerical Crossover under Phase Two*

In this section, we evaluate the performance of the proposed optimization in a context of network and fractal WSN optimization, and then we compare the performance of OC and other types of crossover's fitness.

As mentioned in Section 7.3, numerical crossover is employed, and the floating points are randomly chosen to represent A or B attitudes. The values used range from 0.1 to 0.9. The parameters used in this section for optimization is dependent on N and L from Equations (27) and (28), respectively, to be substituted in Equations (21) and (22). The optimization in these equations is used to get minimal value for parameters in Equations (21) and (22), thus accomplishing the minimization task for the system.

Figures 20 and 21 show the fitness optimization comparison and crossover implementation using Equation (21). It is shown that OC gradually performs the minimization, and clearly outperforms UC. Although OC comes second as compared to NC, OC is still more suitable practically because it considers the fractal geometry which needs gradual optimization through generations (where each level of node is equivalent to a generation) to prevent the communication overhead. On the other hand, QA gave better performance than OC only in the last generation. This means that QA could give maximization at any level of generations, and this behavior does not fit in networks and communications.

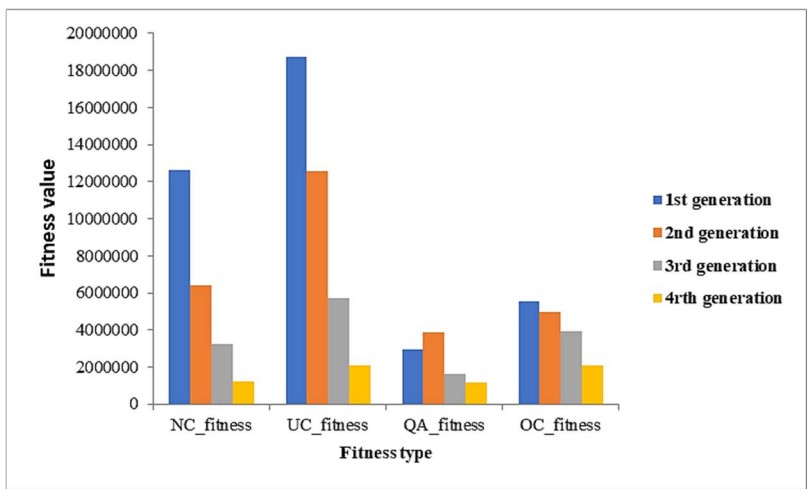

**Figure 20.** Fitness of crossover's types under numerical crossover using Equation (21).

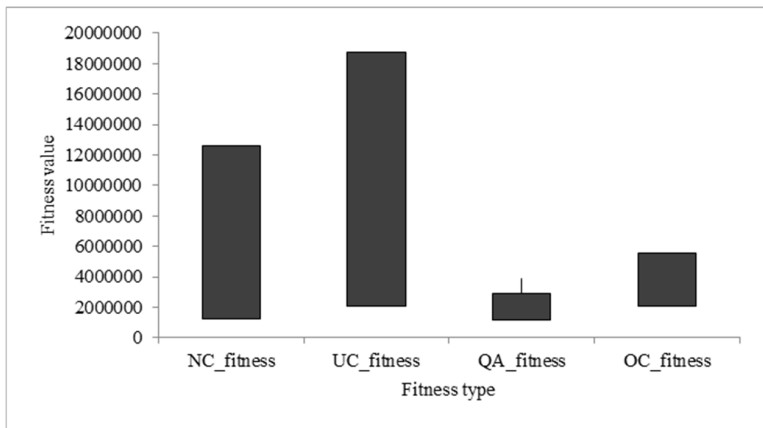

**Figure 21.** Fitness comparison of crossover's types under numerical crossover using Equation (21) sampled by Japan's Candles. The black color for candles concerns decreasing in value.

Figures 22 and 23 show the comparisons of fitness optimization and crossover implementation using Equation (22). It is indicated that the OC's performance decreases gradually among generations. In addition, the fitness gap between the first generation and the last one is small in OC because the system used fractal geometry, and this entails low communication overhead. Hence, OC outperforms UC.

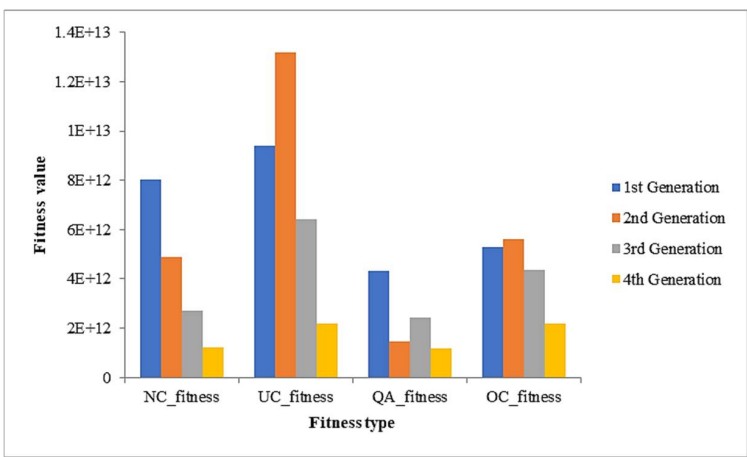

**Figure 22.** Fitness comparison of crossover r's types under numerical crossover using Equation (22).

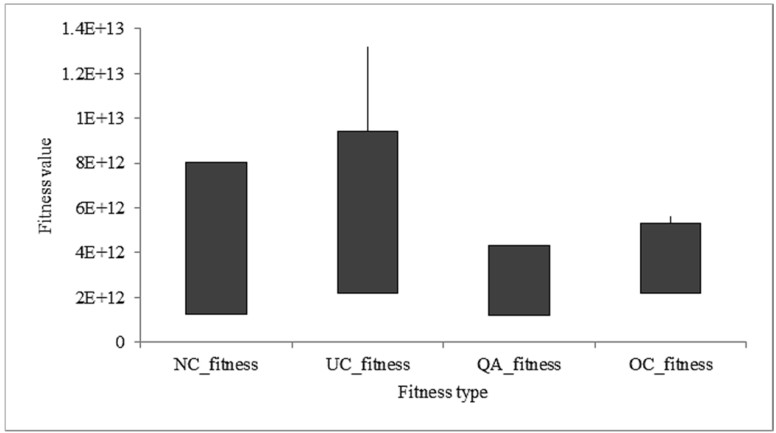

**Figure 23.** Fitness comparison of crossover's types under numerical crossover using Equation (22) sampled by Japan's Candles. Black color for candles refers to decreasing in value.

### 7.7. Equilibrium State under Phase Two

This section presents the experimental search for the best optimization after one hundred iterations for the four types of crossover, with four generations from the selected population for every iteration. Figure 24 shows the counts of iterations to reach the equilibrium state using Equation (19). Figure 25 shows the equilibrium state using Equation (21). The implementation of Equation (22) is shown in Figure 26. Figure 24 shows that OC reaches to the constant state after 40 iterations, but the others have no clear behavior. Figure 25 shows that OC reaches to the equilibrium state after five iterations, faster than UC, but there is no clear behavior for the others. In Figure 26, QA looks like the initial state, but OC has normal behavior and reaches the constant state after 40 iterations.

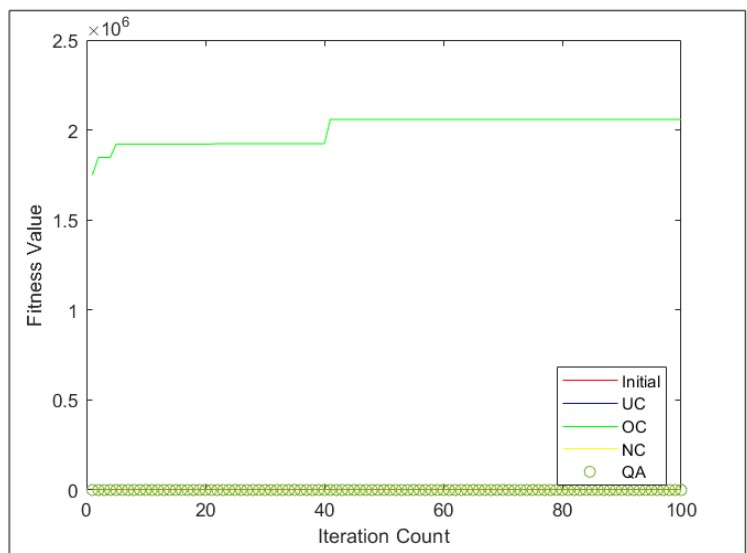

**Figure 24.** Equilibrium state using Equation (19).

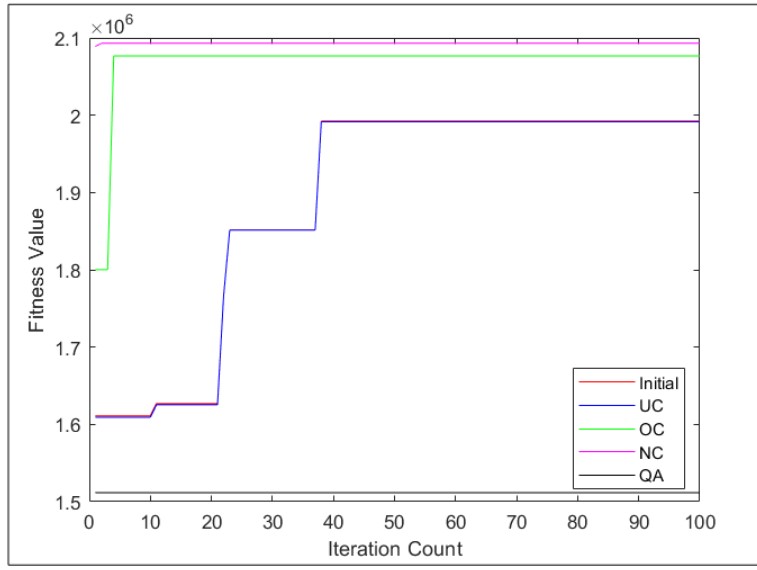

**Figure 25.** Equilibrium state using Equation (21).

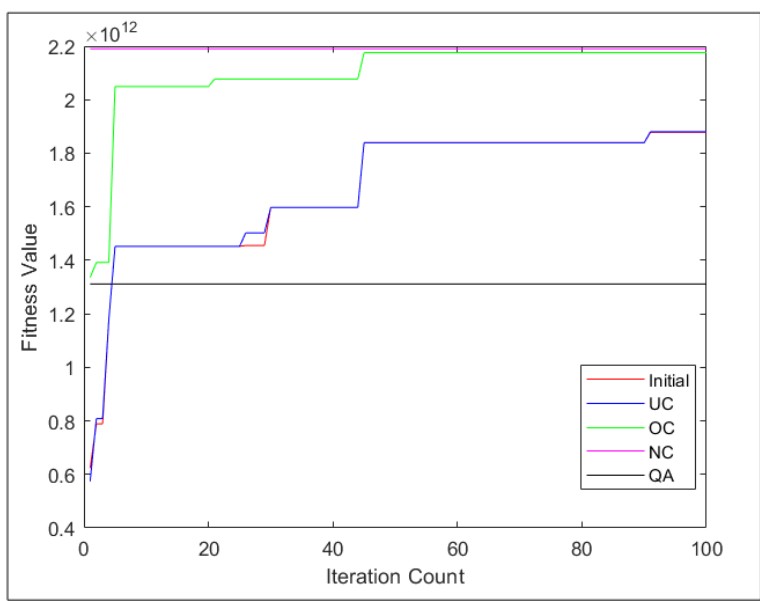

**Figure 26.** Equilibrium state using Equation (22).

## 8. Concluding Remarks

This work presents a new optimization method for genetic algorithms. It is shown that using two properties (or attitudes) in the allele of the genetic algorithm when used to optimize any system (like computer networks, routing algorithms, and others), then the system will be more stable, especially when the two properties are negatively correlated (i.e., when one of them is increased, the other is decreased). The proposed optimization could improve the performance of any system, especially if the system needs to improve some preferred parameters at the price of the unpreferred parameters.

Experimentally, the proposed approach is more suitable for numerical optimization in systems that involve numerical equations and parameters (such as industrial control systems); where experiments have shown that the oriented crossover (OC) outperforms the uniform crossover (UC), NC, and many cases of quarterly crossover (QA). In addition, the experiments have shown that OC is more convenient for the purpose of optimization in networks than UC.

It is expected that the proposed work would be useful in many disciplines, particularly in network and routing techniques.

**Author Contributions:** Conceptualization, F.R. and Z.M.H.; methodology, F.R. and Z.M.H.; software, F.R. and Z.M.H.; validation, F.R. and Z.M.H.; formal analysis, F.R. and Z.M.H. investigation, F.R. and Z.M.H.; resources, Z.M.H.; data curation, F.R. and Z.M.H.; writing—original draft, F.R.; writing—review and editing, Z.M.H.; visualization, F.R. and Z.M.H.; supervision, Z.M.H.; project administration, Z.M.H. All authors have read and agreed to the published version of the manuscript.

**Funding:** This project received internal funding from the University of Kufa.

**Data Availability Statement:** All the types of data were generated using mathematical equations.

**Acknowledgments:** The authors would like to thank the (anonymous) reviewers for their constructive comments that improved this paper.

**Conflicts of Interest:** The authors declare that no conflicts of interest is associated with this work.

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
