# Peer review of "Oriented Crossover in Genetic Algorithms for Computer Networks Optimization"

_information, doi:10.3390/info14050276_

Round 1
Reviewer 1 Report
1. The contribution is not clear enough, the author should mention this in the Introduction.
2. The Introduction and related Works should be enhanced, especially on relative algorithm and their applications. These articles may be helpful for improving this paper: Simulated Annealing Genetic Algorithm Based Schedule Risk Management of IT Outsourcing Project, Genetic Algorithm: Review and Application, Two-Level Principal-Agent Model for Schedule Risk Control of IT Outsourcing Project Based on Genetic Algorithm, Grammar induction using bit masking oriented genetic algorithm and comparative analysis, a hybrid metaheuristic algorithm for the multi-objective location-routing problem in the early post-disaster stage.
3. It is a confusing format for the formula markers involved in the text. It is suggested to align right.
4. It is confusion in paragraph formatting.
5. In section 6.4, it is recommended that specific formula information should not appear in the flow of the subsection. The formula situation should be introduced before the process is completed.
6. It is suggested that the numerical examples used in the experiments and the sources should be clearly stated.
7. One of the purpose of the paper is to design a strong GA, it is recommended to increase the number of algorithm comparisons. It should be compared with other outstanding improved genetic algorithms and other metaheuristics algorithms, like PSO, TS, SA, ACO, et, al.
8. Quality of figures should be improved, such as fig. 17-23
9. How to verify the value the proposed method for the computer networks optimization? I suggest the authors to make more discussion and analysis on problem solving.
Moderate editing of English language
Reviewer 2 Report
Section 2 is interesting but would benefit from a little less stating facts but more on their relevance. Linking or comparative analysis of the techniques a little more.
Do like the use of supposed in this context - perhaps aims?
Section 5.1 is not clear are both cross-overs used together.
Section 6.3 explain the significance of the results
Not sure what figure 16 shows the vertical axis labels are missing.
Just needs a bit of proof-reading make the message clearer.
